# TARPA: Retrieval Priors for Time-Series Foundation Models on Audience Retention

**Dmitriy Petukhov** [1]  **Nikolai Kharchevnikov**  **Leon Useinov** [1]  **Sviatoslav Stumpf** [1]  **Valeria Efimova** [1]

## Abstract

Frozen time-series foundation models (TS-FMs) promise off-the-shelf representations for forecasting tasks with little supervision. TS-FMs are pretrained to forecast future values from past observations. It remains unclear whether this knowledge transfers to predicting retention curves from multimodal content in a few-shot setting. We study this setting on a multimodal structured-output regression task: audience retention prediction on YouTube, with audio, visual, and text-derived per-second channels alongside tabular video-level metadata. We find two paired results — naive featurising fails, prior-conditioning succeeds: (1) A naive frozen TS-FM that consumes the multimodal per-second features does not improve over a no-FM baseline: the TS-FM was pretrained on raw univariate time series, not on engineered features. (2) Conditioning the same TS-FM on a non-parametric retention prior significantly improves how closely the predicted curve matches the true retention shape. Our method, TARPA (Template-Augmented Retrieval Prior Adapter), feeds the retrieval prior as an additional input channel to the frozen TTM backbone, giving the FM at least one channel in the format it was pretrained on. TARPA achieves the best RMSE and Spike-RMSE across all evaluated methods — a tree-based CatBoost regressor, FM-based predictors (TTM, Chronos-Bolt), and FM-free predictors. On this multimodal structured task, TS-FMs yield measurable gains when at least one input channel matches their pretraining format.

[1]ITMO University, Saint Petersburg, Russia. Correspondence to: Dmitriy Petukhov <367479@niuitmo.ru>, Nikolai Kharchevnikov <ask@nikolaichaze.com>, Leon Useinov <370110@edu.itmo.ru>, Sviatoslav Stumpf <sastumpf@itmo.ru>, Valeria Efimova <vefimova@itmo.ru>.

*Proceedings of the $43^{rd}$ International Conference on Machine Learning*, Seoul, South Korea. PMLR 306, 2026. Copyright 2026 by the author(s).

## 1. Introduction

Time series foundation models (TS-FMs) such as TTM (Ekambaram et al., 2024), Chronos-Bolt (Ansari et al., 2024), TimesFM (Das et al., 2024) and Moirai (Woo et al., 2024) are pretrained for forecasting on raw real-valued time series. Audience-retention curves are time series too, but unusual ones: predicted from content rather than extrapolated from the past. Whether TS-FMs, pretrained for the latter, transfer to the former remains an open question. A wave of decoder- and encoder-style TS-FMs has emerged in recent years: TTM (our backbone) is an encoder-decoder based on the MLP-Mixer architecture (Tolstikhin et al., 2021); Chronos (with the faster Chronos-Bolt variant) casts forecasting as language modeling over quantized tokens; TimesFM is a decoder-only forecaster; Moirai unifies multivariate forecasting through a masked encoder. All four share patch-based input encoding rooted in PatchTST (Nie et al., 2023). These models are typically evaluated as forecasters; their use as feature extractors for non-forecasting downstream tasks is far less studied.

The retention curve $y \in [0, 1]^P$ samples the fraction of viewers still watching at moment $P{=}50$ — equally-spaced positions along the video. The curve has a characteristic structure: typically monotonically decreasing, with abrupt drops at sponsored segments. The task is natively multimodal: inputs combine per-second multivariate time-series features (audio, visual, linguistic, content) with tabular video-level metadata and a pooled transcript embedding. The target is itself a fixed-length curve — a multimodal structured-output regression problem in which the prediction is a curve rather than a scalar. Pre-publication prediction is of practical value to creators: edits before release are still possible, and retention drives YouTube's recommendation ranking (Covington et al., 2016), so improving the predicted curve translates directly into a larger audience. At the same time, data is scarce: per-video ground-truth requires creator-shared YouTube Studio analytics, so corpus size rarely reaches $N{\sim}100$.

Most prior work predicts scalar engagement proxies (Trzciński & Rokita, 2017; Bielski & Trzciński, 2018; Chen et al., 2016) or treats engagement and popularity as separable phenomena (Wu et al., 2018). The papers (Stap-

pen et al., 2022) and (Li et al., 2024) target temporal engagement signals rather than full retention curves; the paper (Lin et al., 2023) uses tree-based progressive regression for short-video watch-time, framed within recommendation ranking rather than full-curve reconstruction. No prior work predicts a full retention curve for long-form videos. Gradient-boosted trees such as CatBoost (Prokhorenkova et al., 2018) are the standard strong baseline for tabular small-$N$ tasks (Grinsztajn et al., 2022); TabPFN (Hollmann et al., 2023) shows that pretrained transformers can match or beat trees on small classification tasks via in-context learning, but the analogous result for structured-output regression is open. Naive TS-FM-as-featuriser approaches have produced mixed results, echoing similar negative findings that pretrained transformers do not consistently dominate (Grinsztajn et al., 2022).

The adapter pattern, originating in NLP (Houlsby et al., 2019) and now widespread through LoRA (Hu et al., 2022), freezes a pretrained backbone and inserts a small trainable module. In contrast, our method introduces no trainable parameters into the FM at all: we adapt the model entirely through its input, supplying a non-parametric retrieval prior as an additional channel alongside the observed ones. The use of nearest-neighbor priors as an additional inductive bias is well-established in NLP through $k$NN-LM (Khandelwal et al., 2020) and RAG (Lewis et al., 2020); we instead feed the retrieval prior as additional input channel to the FM, rather than only blending it with predictions post-hoc (as in $k$NN-LM).

Each of these threads — TS-FMs as feature extractors, adapter-based parameter-efficient adaptation, retrieval-augmented prediction, and engagement modeling — is actively developed in isolation. We combine them for a setting not previously addressed: zero-shot full retention curve forecasting from pre-publication content.

We introduce the TARPA model: for each held-out video, it retrieves the mean retention curve of its $k$ nearest training neighbors in metadata space and feeds it to a frozen TS-FM as an extra input channel. This channel is itself a real retention curve — in the format the FM was pretrained on. A trainable predictor combines the FM's output with the per-second observations to yield the predicted curve.

## 2. Method

Figure 1 sketches TARPA's pipeline: for each held-out video, a non-parametric retrieval prior — the mean retention curve of its $k$ nearest training neighbors by metadata —— is fed as an additional input channel to a frozen TS-FM, whose per-channel encodings are combined with video-level metadata by a trainable predictor that produces the final retention curve. We describe each component in turn; the loss and

training details are deferred to Section A.

### 2.1. Setup

For each video $v$, we are given a per-second observation matrix $X_v \in \mathbb{R}^{T_v \times C}$ with $C{=}75$ engineered audio/visual/content channels (text-derived channels from a Whisper (Radford et al., 2023) transcript), a video-level vector $z_v \in \mathbb{R}^{38}$ summarising channel metadata and aggregated LLM-derived statistics, and a pooled transcript-segment embedding $s_v \in \mathbb{R}^{256}$. The target is the retention curve $y_v \in [0,1]^P$. We train under leave-one-out cross-validation (LOO), which preserves the maximum training signal per fold and yields paired per-video predictions for downstream statistical testing.

### 2.2. TS-FM backbone

Our TS-FM is TTM, a pretrained MLP-Mixer-based encoder for univariate forecasting, with weights frozen throughout. Following the patch-based input encoding inherited from PatchTST (Nie et al., 2023), each of the $C$ channels is processed univariately: resampled to a fixed 512-point context via linear interpolation and split into $P_p{=}8$ non-overlapping patches of 64 points; the FM emits a $d{=}192$-dimensional embedding per patch. This yields the per-channel hidden-state cube $H_v \in \mathbb{R}^{C \times P_p \times d}$.

### 2.3. Predictor architecture

The predictor (Figure 2) builds the predicted retention curve from three complementary components. Curve shape is parameterized by functional-PCA (FPCA) coefficients $c_v \in \mathbb{R}^K$ on a precomputed basis $B$ capturing the dominant modes of training-curve variation; absolute level is set by a scale-and-shift pair $(s_v^{\text{lvl}}, b_v^{\text{lvl}})$, zero-initialized so the predictor begins at identity and deviates only when the data demands it; and a small bounded per-position residual $\delta_v \in [\pm 0.05]^P$ permits local corrections beyond what the FPCA backbone can express. We combine these components into the predicted curve as

$$\hat{y}_v = \text{clip}\big((\bar{y} + B^\top c_v)\, s_v^{\text{lvl}} + b_v^{\text{lvl}} + \delta_v - \pi_v\big), \quad (1)$$

where $\text{clip}(\cdot)$ projects componentwise onto $[0,1]$, $\bar{y}$ is the mean training curve, and $B \in \mathbb{R}^{K \times P}$ (Ramsay & Silverman, 2005) ($K{=}6$ basis functions, capturing >97% of training-curve variance) is precomputed from the training set.

A shared trainable backbone produces $(c_v, s_v^{\text{lvl}}, b_v^{\text{lvl}}, \delta_v)$ from the per-channel TTM encodings $H_v$ together with the video metadata $z_v, s_v$. It pools the $(C{+}1)$ channels of $H_v$ into a $P$-position sequence with a cross-channel attention (learned per-position queries, per-channel bias), fuses this with linearly-projected per-position text features, and contextualizes the positions with a small pre-LN Transformer. Three

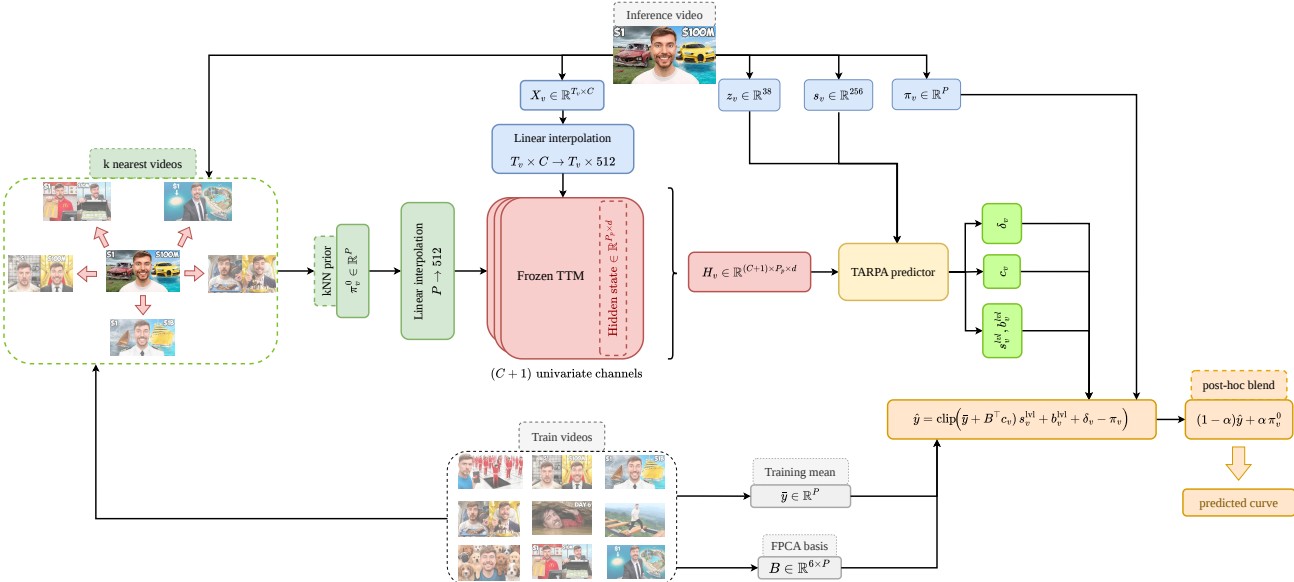

*Figure 1.* **TARPA pipeline.** A retrieval prior — the mean retention curve of the nearest training videos by metadata distance — is concatenated with the per-second engineered features and fed to the frozen TTM as an extra input channel. The TARPA predictor combines TTM encodings with video-level metadata to produce shape, level, and residual parameters, which compose into the predicted curve via Equation (1). An optional post-hoc blend with the retrieval prior yields the final output.

small MLP heads on the contextualized positions read out $c_v$, $(s_v^{\mathrm{lvl}}, b_v^{\mathrm{lvl}})$, and $\delta_v$ respectively.

The integration penalty $\pi_v$ is a deterministic per-position correction over the sponsor segment of $v$ (text-extracted timestamps). It is inherited unchanged from the CatBoost integration-penalty baseline and applied identically across all predictor variants. In our Predictor w/o FM ablation, the TS-FM is removed: each channel of $X_v$ is split into $P_p$ time patches and averaged, yielding a $\mathbb{R}^{C \times P_p \times 1}$ tensor that replaces $H_v$ in the predictor pipeline.

### 2.4. TARPA: prior-conditioned input

For a held-out video $v$, TARPA augments the TS-FM input with a $(C{+}1)$-th retention prior channel

$$\pi_v^0 \;=\; \tfrac{1}{k} \sum_{u \in \mathcal{N}_k(v)} y_u, \qquad (2)$$

where $\mathcal{N}_k(v)$ are the $k{=}5$ nearest training videos to $v$ by Euclidean distance in standardized $z$-space. Unlike the engineered channels in $X_v$, this prior is a real $[0,1]^P$ retention curve — a smooth, bounded 1-D time series in the exact format the TS-FM was pretrained on. We resample $\pi_v^0$ to the FM context and concatenate it with the $C$ observation channels; the FM runs on the $(C{+}1)$-channel input, producing per-channel encodings $H_v \in \mathbb{R}^{(C+1) \times P_p \times d}$ whose last slice encodes the prior. The predictor then composes $\hat{y}_v$ from $H_v$ and $(z_v, s_v)$ via Equation (1), unchanged from the no-prior variants.

### 2.5. Post-hoc blend

We optionally blend the predictor's output with the initial prior, following $k$NN-LM (Khandelwal et al., 2020):

$$\hat{y}_v \leftarrow (1-\alpha)\hat{y}_v + \alpha\,\pi_v^0, \quad \alpha = 0.10. \qquad (3)$$

### 2.6. Loss and training

The composite loss combines pointwise RMSE with shape, smoothness, and level auxiliaries; we train for 200 epochs at batch 8 with AdamW, mixup (Zhang et al., 2018) ($\alpha{=}0.2$), SWA (Izmailov et al., 2018) over the last 40% epochs, and light input noise. Full loss decomposition, hyperparameters, and the predictor-internals diagram are in Section A.

## 3. Experiments

We collected a corpus of $N{=}85$ long-form videos from a single YouTube channel (durations 5.5-33 min); inputs and target follow the notation of Section 2. Sponsor presence in $X_v$ is detected via SponsorBlock (Ramachandran, 2023); the text-derived channels comprise interval-segment labels produced by prompting gpt-4.1-mini on the Whisper transcript, together with deterministic linguistic/affect statistics from the same transcript; targets $y_v$ are YouTube Studio retention curves.

Strict LOO with a light predictor ($\approx$96k trainable parameters; the TTM backbone is frozen). We report five metrics: SRCC (Spearman rank correlation between $\hat{y}_v$ and $y_v$) and PLCC (Pearson linear correlation) measure shape fidelity;

*Table 1.* **Leave-one-out cross-validation results.** Mean of each metric across all videos. **Bold** / underline = best / second-best per column. ↑/↓: higher/lower is better. Method categories are described in the *Baselines* paragraph above; paired bootstrap CIs vs. TARPA are in Table 2 (appendix).

| Method | SRCC↑ | PLCC↑ | RMSE↓ | MAE↓ | Spike-RMSE↓ |
|---|---|---|---|---|---|
| Global Mean Baseline | 0.9199 | 0.9214 | 0.0870 | 0.0789 | 0.0245 |
| CatBoost Integration Penalty | 0.9375 | 0.9334 | 0.0784 | **0.0704** | 0.0253 |
| TTM MLP Probe | 0.9096 | 0.9176 | 0.0840 | 0.0763 | 0.0249 |
| Raw predictor + blend | 0.9518 | 0.9413 | 0.0777 | 0.0705 | 0.0238 |
| TTM predictor | 0.9531 | 0.9424 | 0.0790 | 0.0719 | 0.0240 |
| Chronos-Bolt predictor | 0.9275 | 0.9260 | 0.0806 | 0.0731 | 0.0246 |
| TARPA w/o FM | 0.9525 | 0.9412 | 0.0806 | 0.0735 | 0.0243 |
| TARPA w/o blend | **0.9567** | **0.9440** | 0.0790 | 0.0719 | 0.0241 |
| **TARPA (ours)** | 0.9560 | 0.9439 | **0.0776** | 0.0706 | **0.0236** |

RMSE and MAE measure absolute level; Spike-RMSE is the RMSE on the first differences $\Delta \hat{y}_v$ vs. $\Delta y_v$, quantifying alignment of abrupt drops (typically at sponsor segments).

Our baselines cover four method categories. *Global Mean* predicts the mean training curve on every video. *CatBoost Integration Penalty* (Prokhorenkova et al., 2018) is a strong tree-based baseline: for each output position $t \in \{1, \ldots, P\}$ we fit a separate gradient-boosted regressor on $z_v$ plus per-second features at that position, and a parallel delta-regressor predicts $y_t - y_{t-1}$ cumulated into a smoothed curve; the two outputs are blended. It uses neither the TS-FM, our trainable predictor, nor the retrieval prior, but the same deterministic sponsor penalty $\pi_v$ is applied post-hoc. *TTM MLP Probe* is a small MLP head with trainable budget matched to our predictor, on top of frozen TTM (Ekambaram et al., 2024) encodings of $X_v$. *Raw predictor + blend*, *TTM predictor*, and *Chronos-Bolt predictor* share our full predictor architecture and differ only in the input source — patch-mean of $X_v$, TTM encodings, or Chronos-Bolt (Ansari et al., 2024) encodings respectively – without the retrieval-prior channel; the first additionally applies the post-hoc blend. *TARPA* adds the retrieval prior as a $(C+1)^{\text{th}}$ input channel to the FM and applies the post-hoc blend; *TARPA w/o FM* and *TARPA w/o blend* drop the FM or the blend respectively.

The results are presented in Tab 1. Naive TS-FM use does not outperform a no-FM predictor. The most striking sign that naive use does not work is that a small MLP probe on frozen TTM encodings performs worse than the trivial Global Mean Baseline on Spearman. Substituting Chronos-Bolt for TTM yields the same negative pattern. Inside our full predictor architecture, TTM only ties the no-FM raw predictor + blend at SRCC 0.953, and Chronos-Bolt falls below it. The FM offers no useful prior over engineered audio/visual channels.

TARPA's predicted curves more closely match the true retention shape. TARPA achieves the best score on RMSE and Spike-RMSE and the second-best on SRCC and PLCC. Against the CatBoost Integration-Penalty baseline, it significantly improves SRCC by 0.019 and Spike-RMSE by 0.0017 while matching RMSE and MAE. Against the strongest non-FM predictor, the raw predictor + blend, TARPA matches RMSE within $10^{-4}$ and still significantly improves SRCC by 0.0042. We attribute these shape gains to the TS-FM specifically: removing only the FM from TARPA, while keeping the prior and the predictor unchanged, costs a significant Spike-RMSE advantage on the metric most directly sensitive to shape, ruling out the prior and the trainable backbone as the source. The gains concentrate in the shape-fidelity metrics, consistent with the FM contributing a structural prior rather than a level correction. This pattern is robust to the blend strength: across $\alpha \in [0, 0.30]$, TARPA lies strictly above the no-FM curve in (SRCC, RMSE) space, so the improvement is not specific to our default $\alpha$.

## 4. Conclusion

The central question for foundation models on structured data is when their pretraining helps with a new task. Our experiments on $N{=}85$ YouTube retention curves identify input-format alignment as a primary axis of that question: a frozen TS-FM used as a feature extractor over engineered audio/visual/content channels offers no improvement over no-FM baselines, while conditioning the same FM on a non-parametric retrieval prior over retention curves (TARPA) — that is, supplying the FM with an input channel in the format it was pretrained on — recovers measurable gains in matching the true retention shape. TARPA achieves the best RMSE and Spike-RMSE among all methods evaluated, and an ablation study shows the TS-FM is what drives these gains. TARPA is evaluated with TTM as the prior-conditioning FM, on a single-creator corpus ($N{=}85$); replication with other TS-FM families and across creators is left to future work.

## Acknowledgments

This work supported by the Ministry of Economic Development of the Russian Federation (IGK 000000C313925P4C0002), agreement No139-15-2025-010.

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

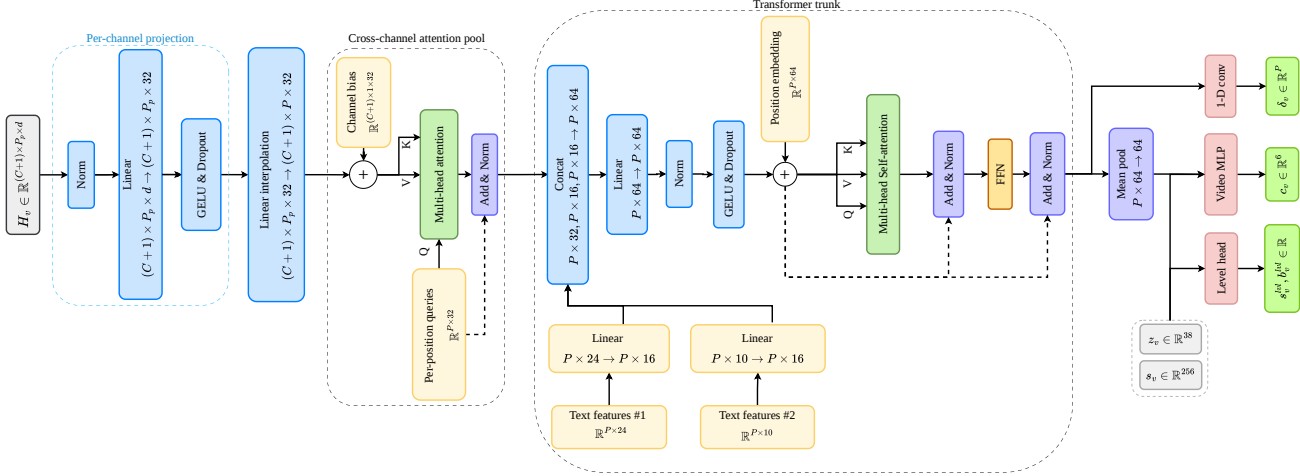

*Figure 2.* **TARPA predictor internals ($\sim$96k trainable parameters).** Per-channel TTM encodings are projected, resampled to the curve length, then collapsed into a per-position sequence via cross-channel attention. The sequence is fused with per-position text features and contextualised through a small Transformer encoder. Three output heads produce the residual, the FPCA shape coefficients, and the level scale/shift; together with the precomputed mean curve and FPCA basis they compose into the final retention curve.

## A. Loss, training, and predictor diagram

**Loss.** For each video we minimise a composite per-video loss $\mathcal{L}_v = \mathrm{RMSE}(\hat{y}_v, y_v) + \sum_i \lambda_i \mathcal{L}_v^{(i)}$, with pointwise RMSE as the dominant term and auxiliaries grouped into three families. *Shape:* a Pearson-correlation gap, first-difference RMSE on $\hat{y}_v$ (spike) and second-difference MSE (curvature) – the correlation term penalises monotonicity violations that RMSE ignores, while the difference-based terms target the abrupt drops characteristic of sponsor segments and the smoothness of the remaining curve. *Smoothness regularisers:* total variation on $\hat{y}_v$ and $L_1$ on $\delta_v$, keeping the bounded residual head near zero so the FPCA backbone $\bar{y} + B^\top c_v$ carries the curve. *Level:* two $L_1$ aggregates – the mean over all $P$ positions of the curve and the mean over the last 5 – capturing the overall retention level and the tail retention. The same loss and weights are used across all predictor variants we compare.

**Training.** We train for 200 epochs at batch size 8 with AdamW (learning rate $1.5\times10^{-3}$, weight decay $5\times10^{-3}$). Regularisation in this small-sample regime combines mixup (Zhang et al., 2018) with $\alpha=0.2$, stochastic weight averaging (Izmailov et al., 2018) over the last $40\%$ of epochs, and light Gaussian input noise; early stopping uses an inner 6-of-84 train/val split. Mixup and SWA are standard small-sample regularisers, chosen to ensure predictor capacity is not the limiting factor when the TS-FM is used as a generic feature extractor. The TS-FM stays frozen throughout, and the trainable surface is identical across the predictor variants we compare.

*Table 2.* **Paired per-video improvement of TARPA over each baseline.** Each cell is the per-video mean of TARPA score $-$ method score $\pm$ the half-width of its bootstrap confidence interval. **Bold**: TARPA significantly wins; otherwise no significant difference.

| Method | SRCC $\uparrow$ | PLCC $\uparrow$ | RMSE $\downarrow$ | MAE $\downarrow$ | Spike-RMSE $\downarrow$ |
|---|---|---|---|---|---|
| vs. CatBoost Int. Penalty | **$0.0186 \pm 0.0077$** | **$0.0105 \pm 0.0050$** | $-0.0007 \pm 0.0065$ | $0.0003 \pm 0.0066$ | **$-0.0017 \pm 0.0006$** |
| vs. Raw + blend | **$0.0042 \pm 0.0041$** | $0.0027 \pm 0.0031$ | $-0.0000 \pm 0.0049$ | $0.0001 \pm 0.0050$ | $-0.0002 \pm 0.0004$ |
| vs. TTM predictor | $0.0029 \pm 0.0045$ | $0.0016 \pm 0.0030$ | $-0.0014 \pm 0.0055$ | $-0.0012 \pm 0.0056$ | $-0.0003 \pm 0.0005$ |
| vs. TARPA w/o FM | $0.0035 \pm 0.0043$ | $0.0028 \pm 0.0034$ | $-0.0030 \pm 0.0052$ | $-0.0028 \pm 0.0054$ | **$-0.0006 \pm 0.0004$** |
| vs. TARPA w/o blend | $-0.0007 \pm 0.0042$ | $-0.0000 \pm 0.0031$ | $-0.0013 \pm 0.0060$ | $-0.0012 \pm 0.0062$ | $-0.0004 \pm 0.0005$ |

