# OpenReview forum: "TARPA: Retrieval Priors for Time-Series Foundation Models on Audience Retention"
_ICML.cc/2026/Workshop/FMSD — FMSD @ ICML 2026 Poster_

### Official Review · Reviewer_o7gv · 2026-05-19
**Clear and very interesting use of retrieval priors for TS-FM adaptation**

**Rating:** 8
**Confidence:** 4

**Review:**

This paper studies whether frozen time-series foundation models can help with YouTube audience-retention curve prediction, a multimodal structured-output regression task where the target is a full retention curve rather than a scalar engagement score. The authors show that naively using a frozen TS-FM over engineered audio/visual/text features does not provide much benefit, because these channels are not in the format the TS-FM was pretrained on. They then propose TARPA, which retrieves the mean retention curve of nearby training videos and feeds this retrieval prior as an additional curve-like input channel to a frozen TTM backbone.

The main strength of the paper is the clean conceptual insight stating that input-format alignment matters when transferring TS-FMs to non-standard structured-output tasks. The result that a TS-FM is not useful when fed arbitrary engineered multimodal features, but becomes useful when given a real-valued retrieval-prior curve, is interesting and highly relevant to foundation models for structured data. The proposed method is also simple and well motivated. The retrieval prior is non-parametric and intuitive, the TTM backbone remains frozen, and the predictor is structured around FPCA shape coefficients, level adjustment, and bounded residual corrections, which is appropriate for the small-data regime.

The empirical evaluation is also fairly strong. TARPA achieves the best RMSE and Spike-RMSE and competitive shape-correlation metrics. I also appreciate the paired bootstrap analysis and the clear discussion that the gains concentrate more in shape-sensitive metrics than in absolute-level metrics.

One main limitation is the size and scope of the dataset. The evaluation uses only 85 videos from a single YouTube channel, so it is hard to know whether TARPA learns a general audience-retention mechanism or a creator-specific curve prior. This is understandable given the difficulty of obtaining YouTube Studio retention curves, but it should make the claims more cautious. Replication across multiple creators, content genres, and channel sizes would substantially strengthen the paper.

A second concern is that the gains, while consistent, are modest. For example, the difference between TARPA and the strongest non-FM predictor is quite small on RMSE and MAE, although TARPA is stronger on shape and spike-sensitive metrics. I would also like to see a pure kNN-prior baseline that predicts only the mean curve of the retrieved neighbors, without the trainable predictor, to isolate how much performance comes from retrieval alone.

Overall, I find this to be a clear, relevant, and technically interesting work. The central input-alignment idea is compelling, the method is simple, and the experiments support the main claim despite the small-data setting.

---

### Official Review · Reviewer_TdfE · 2026-05-20

**Rating:** 5
**Confidence:** 3

**Review:**

- Summary: The paper studies audience-retention curve prediction for YouTube videos. The input combines per-second audio/visual/text-derived channels, video-level metadata, and transcript embeddings. The target is a fixed-length retention curve. The key observation is that using a frozen time-series foundation model directly on engineered multimodal channels does not help, because the TS-FM was pretrained on raw univariate time series rather than engineered feature streams. The proposed method, TARPA, instead retrieves the mean retention curve of the k=5 nearest training videos in metadata space and feeds that retrieval prior as an additional input channel to a frozen TTM backbone. The method then combines frozen TTM encodings with metadata using a lightweight predictor to produce FPCA shape coefficients, level scale/shift, and local residual corrections.

- Strengths: The paper asks a relevant FMSD question: when does a pretrained time-series foundation model help on a downstream structured task that is not standard forecasting? The answer proposed here, input-format alignment matters, is plausible and useful. The task is interesting. Predicting an entire retention curve is more structured than scalar engagement prediction, and the few-shot setting is realistic because YouTube Studio retention curves are hard to collect.

- Weaknesses and Questions: The main idea built from retrieval has been well discovered in time series predictions, so the novelty is very limited. Also the paper evaluate the method on a very small and narrow datasets with n=85 videos from only one single Youtube channel. Also, the gains are modest. TARPA improves over CatBoost on SRCC and Spike-RMSE, but RMSE/MAE are essentially similar. Compared with the strongest no-FM predictor, the improvement is mostly in shape fidelity, and the absolute gains are small. In addition, The retrieval prior itself is very strong and may explain much of the performance. The paper argues that removing TTM from TARPA hurts Spike-RMSE, but TARPA w/o FM is still quite competitive. More ablations are needed to isolate whether the FM is learning useful structure from the prior channel or whether the predictor is mostly regularized by a good retrieved template.

---

### Official Review · Reviewer_Kibv · 2026-05-21
**Review of TARPA**

**Rating:** 5
**Confidence:** 4

**Review:**

## Summary
TARPA proposes conditioning a frozen time-series foundation model (TTM) on the mean retention curve of k-nearest training neighbors as an additional input channel, enabling it to predict Youtube audience retention curves for pre-publication Youtube videos. The core insight is that input-format alignment matters: naive use of TS-FMs over engineered features fails, but feeding the FM a channel that matches its pretraining format (a real retention curve) recovers measurable gains in shape fidelity.

## Strengths

### 1. Clean insight:
The paper's central claim that input-format alignment is the key axis determining whether a frozen TS-FM helps is clearly stated and directly tested.

### 2. Novel task formulation.
Framing audience retention prediction as a structured-output regression problem is a meaningful contribution over prior work, which largely predicted scalar engagement proxies.

## Weaknesses

### 1. Severely limited dataset scale and scope:
N=85 videos from a single YouTube channel is a very narrow empirical base. All results, including confidence intervals, are conditioned on one creator's style, topic distribution, and audience. Generalisability to other creators, genres, or platforms is entirely untested, and the authors acknowledge this only briefly in the conclusion.

### 2. Retrieval prior may be doing most of the work:
The ablation shows TARPA w/o FM already achieves competitive SRCC (0.9525) and RMSE (0.0806), while full TARPA's gains over the no-FM baseline are modest and sometimes within confidence intervals (Table 2). It is not fully clear whether the FM adds value beyond what the retrieval prior alone provides, or whether a simpler model using only the prior would suffice.

## Questions for Authors

1. N=85 videos from a single channel means training and test distributions are very similar. Have you considered whether results would hold under a harder split on held-out topic clusters?

2. The TARPA w/o FM variant already performs strongly. Can you provide a direct comparison against a pure kNN prior (no FM, no trainable predictor) to establish the minimum baseline the FM needs to beat?